# Internet Altruistic Behaviors in Adolescents: Roles of Attention-Deficit/Hyperactivity Disorder, Impulsiveness, and Perceived Social Support

**DOI:** 10.3390/bs14060433

**Published:** 2024-05-22

**Authors:** Pin-Han Peng, Yi-Lung Chen, Ray C. Hsiao, Cheng-Fang Yen, Wen-Jiun Chou

**Affiliations:** 1Department of Child and Adolescent Psychiatry, Chang Gung Memorial Hospital, Kaohsiung Medical Center, Kaohsiung 83301, Taiwan; pph840703@cgmh.org.tw; 2Department of Healthcare Administration, Asia University, Taichung 41354, Taiwan; 3Department of Psychology, Asia University, Taichung 41354, Taiwan; 4Department of Psychiatry and Behavioral Sciences, University of Washington School of Medicine, Seattle, WA 98195, USA; 5Department of Psychiatry, Children’s Hospital and Regional Medical Center, Seattle, WA 98105, USA; 6Department of Psychiatry, Kaohsiung Medical University Hospital, Kaohsiung 80708, Taiwan; 7Department of Psychiatry, School of Medicine, College of Medicine, Kaohsiung Medical University, Kaohsiung 80708, Taiwan; 8College of Professional Studies, National Pingtung University of Science and Technology, Pingtung 91201, Taiwan; 9College of Medicine, Chang Gung University, Taoyuan City 33302, Taiwan

**Keywords:** Internet altruistic behavior, attention-deficit/hyperactivity disorder, impulsivity, social support, adolescent, psychological well-being

## Abstract

This study examined the associations of an attention-deficit/hyperactivity disorder (ADHD) diagnosis, impulsivity, and perceived social support with Internet altruistic behaviors (IABs) in adolescents and the associations of IABs with depression, suicidality, and nonsuicidal self-injury in this group. In total, 176 adolescents aged between 11 and 18 years with ADHD and 173 adolescents without ADHD (matched with the ADHD group by sex and age) participated in this study. The adolescents rated their IABs on the Internet altruistic behavior scale, impulsivity on the Barratt impulsiveness scale version 11, and perceived family and peer support on the family and social relationship domains of the Taiwanese quality of life questionnaire for adolescents. The associations of ADHD, impulsivity, and social support with IABs and the associations of IABs with depression, suicidality, and nonsuicidal self-injury were examined through multivariable linear regression analysis. The present study found that more time spent on the Internet (*p* < 0.001), greater perceived peer support (*p* < 0.001), greater impulsiveness characterized by a lack of self-control and perseverance (*p* < 0.001), poorer ability to plan and look ahead (*p* < 0.001), and an ADHD diagnosis (*p* = 0.003) were significantly associated with a higher level of IABs. IABs were not significantly associated with severe depression, suicidality, or nonsuicidal self-injury (all *p* > 0.05). The results of this study indicated that multiple individual and social factors were associated with IABs in adolescents. IABs were not significantly associated with severe depression, suicidality, or nonsuicidal self-injury in adolescents.

## 1. Introduction

Internet altruistic behaviors (IABs) involve actions performed voluntarily to help others on the Internet without any expectation of receiving benefits and in the absence of external pressure from others [1]. According to Zheng [2], common IABs include Internet support (e.g., giving attention to and encouraging netizens), Internet guidance (e.g., guiding netizens on how to use the Internet more effectively), Internet sharing (e.g., sharing your success with others online), and Internet reminders (e.g., warning netizens about network traps). Empirical studies have also demonstrated that face-to-face altruistic behavior is positively associated with engagement in IABs, supporting the argument that IABs and physical-world altruistic behaviors are similar in nature [3]; however, the anonymity and accessibility of the Internet create a particularly conducive environment for IABs [4,5]. The Internet is a key living space for adolescents in modern times. Most studies have focused on how Internet use negatively affects adolescents’ behaviors [6], mental health [7], visual acuity [8], and sleep quality [9]; however, the Internet is also a space where adolescents can explore themselves, develop their self-identity, and enhance their peer support [10]. Examining adolescent IABs can provide insights into the potentially positive aspects of Internet use.

Several topics pertaining to adolescent IABs warrant further investigation. First, most studies have examined the IABs of college students [1,3,5,11,12], with only one study examining the IABs of middle school students [13]. Notably, no study has examined the IABs of adolescents with a neurodevelopmental disease, such as attention-deficit/hyperactivity disorder (ADHD). A meta-analysis reported that 5.6% of adolescents aged 12 to 18 years have ADHD [14], and the predominant symptom has been reported to be inattentiveness, hyperactivity, and impulsiveness [15]. Adolescents with ADHD develop impairments in multiple dimensions of functioning; for example, they may have poor social relationships, low self-esteem, and poor academic performance [16]. According to ecological systems theory [17], IABs result from the interactions between individuals and their environment. Given that adolescents with ADHD exhibit the psychological characteristics of boredom proneness, rapid habituation to repeated positive reinforcement, and a high propensity for seeking fun, the Internet provides a conducive space for them to pursue pleasurable experiences [18,19]. The Internet also gives adolescents with ADHD opportunities to develop social relationships and gain peer recognition, which are more difficult for them to achieve through face-to-face interactions [20]. Furthermore, Internet gaming is an important leisure activity for adolescents with ADHD [21], and the Internet offers numerous opportunities to share gaming tips with others. Studies have demonstrated that ADHD is one of the risk factors for developing problematic Internet use [22,23,24]. Studies have also focused on the negative effects of problematic Internet use on mental health and social skills among adolescents [25,26]; however, the IABs of adolescents with ADHD remain unexplored. Thus, the following hypothesis was proposed:

**H1:** 
*Adolescents with ADHD exhibit higher levels of IABs relative to those without ADHD.*


Second, scholars have proposed that physical-world altruistic behaviors are the product of rational thinking [27] and inner responsibility [28]. Comparatively, the requirements for engaging in online IABs are less strict than those for engaging in physical-world altruistic behaviors; therefore, individuals are more likely to engage in online IABs without careful consideration [29,30]. Whether the psychological trait of impulsiveness contributes to adolescents’ IABs is unclear. Impulsiveness is characterized by the underestimation of harm, nonreflective responses, difficult-to-control desires, and repetitive pleasure- as well as gratification-seeking behaviors [31]. According to Barratt [32], impulsiveness has three dimensions, namely the inability to plan and look ahead, a lack of self-control and perseverance, and novelty-seeking behavior. Given that impulsivity is a core symptom of ADHD, whether ADHD moderates or mediates the association between impulsiveness and adolescent IABs must be clarified. Thus, the following hypotheses were proposed:

**H2a:** 
*Impulsiveness is significantly associated with IABs in adolescents.*


**H2b:** 
*ADHD moderates the association between impulsiveness and IABs in adolescents.*


**H2c:** 
*ADHD mediates the association between impulsiveness and IABs in adolescents.*


Third, families and peers are important sources of support for adolescents. Studies have reported that parental emotional warmth [12,33] and peer support [11] are positively associated with IABs. Some scholars hypothesized that individuals experience active emotions when they receive social support from their environment and are then willing to engage in altruistic behaviors [11]; however, adolescents with ADHD experience more conflict with their parents [34] and peer rejection [35] relative to those without ADHD. Thus, the relationship between perceived social support and IABs may be different for adolescents with ADHD versus those without ADHD. Given these literature findings, the following hypotheses were proposed:

**H3a:** 
*Perceived family support is significantly associated with IABs in adolescents.*


**H3b:** 
*Perceived peer support is significantly associated with IABs in adolescents.*


**H3c:** 
*ADHD moderates the associations between perceived family support and IABs in adolescents.*


**H3d:** 
*ADHD moderates the associations between perceived peer support and IABs in adolescents.*


Fourth, studies have discovered that both physical-world altruistic behaviors [36,37,38] and IABs [5,13] have positive associations with mental health. Altruistic behaviors have been proposed to help individuals recognize their self-worth and experience active emotions, thereby improving their mental health [33]; however, individuals with ADHD have higher risks of comorbid depression [39] and suicidality [40] compared with those without ADHD. Thus, whether the association between IABs and mental health in adolescents is significant after adjusting for ADHD must be clarified. To this end, the following hypothesis was proposed:

**H4:** 
*In adolescents, IABs are significantly associated with reduced risks of depression, suicidality, and nonsuicidal self-injury.*


The objectives of the present study were to examine the associations of ADHD, impulsivity, and perceived social support with IABs in adolescents as well as the associations of IABs with adolescents’ depression, suicidality, and nonsuicidal self-injury. The moderating effect of ADHD on the associations of IABs with other factors was also examined. Given that prolonged Internet use increases the likelihood of engagement in IABs, the present study adjusted for the effect of time spent on the Internet in its analysis.

## 2. Methods

### 2.1. Participants

The study participants comprised adolescents with ADHD recruited from six psychiatry outpatient clinics of two hospital-affiliated hospitals in Taiwan. All adolescents must meet the following intake criteria: (1) between 11 and 18 years of age and (2) having ADHD diagnosed by a certified child psychiatrist based on the criteria of the *Diagnostic and Statistical Manual of Mental Disorders, fifth edition* (*DSM-5)* [41]. The recruitment of participants was conducted between 1 September 2022 and 31 July 2023. Subsequently, researchers consecutively approached 190 adolescents with ADHD who visited the outpatient clinics. Child psychiatrists interviewed the adolescents with ADHD and their parents (or guardians) to ascertain that they met the aforementioned criteria as well as exclude adolescents who had comorbid intellectual disability, autism spectrum disorder, major depressive disease, bipolar I disorder, schizophrenia, or any other cognitive dysfunction that could impede their understanding of the study’s purposes and completion of the research questionnaire. Finally, 14 adolescents with ADHD were excluded because they had comorbid autism spectrum disorder (*n* = 8), intellectual disability (*n* = 5), or major depressive disorder (*n* = 1). The researcher detailed the purpose and steps of the study and invited the adolescents to participate in the study. The researcher also guaranteed that the adolescents’ information and questionnaire responses will be kept confidential and that participation in the study will not affect their right to medical care. All 176 adolescents with ADHD agreed to participate and completed all questionnaires.

Typically developing (TD) adolescents were recruited through online advertising. The inclusion criteria were as follows: (1) age of 11–18 years and (2) no diagnosis of ADHD, ASD, intellectual disability, autism spectrum disorder, major depressive disease, bipolar disorder, schizophrenia, or any other cognitive deficit that could impede their understanding of the study’s purposes and completion of the research questionnaire. Child psychiatrists interviewed the adolescents and their parents (or guardians) to confirm that the adolescents met the inclusion criteria. Finally, 173 TD adolescents agreed to participate in the present study. Both the adolescents and their parents provided written informed consent prior to the commencement of assessments. The protocol for the present study was approved by the institutional review boards of Kaohsiung Medical University (KMUHIRB-SV(I)-20200091) and Chang Gung Memorial Hospital, Kaohsiung Medical Center (202101964A3C502).

### 2.2. Measures

#### 2.2.1. Internet Altruistic Behavior Scale

We used the 26-item Internet altruistic behavior scale (IABS) to measure the self-reported IABs of the participants [1]. The IABS comprises four dimensions, namely Internet support, Internet guidance, Internet sharing, and Internet reminders. The items of the scale were rated on a 4-point scale with endpoints ranging from 1 (*never*) to 4 (*always*). A higher score indicates a higher level of IABs. Several studies have indicated that the IABS exhibits good reliability and validity [1,2]. In the present study, the IABS had a Cronbach’s α of 0.93, and its subscales had Cronbach’s α values ranging from 0.73 to 0.88. Because of the high correlation between subscale scores (Pearson’s correlation coefficients: 0.526~0.741), the scores for the four subscales of the IABS were transformed through factor analysis (principal component method) to extract a factor score that represented IABs.

#### 2.2.2. Barratt Impulsiveness Scale Version 11

The 25-item Mandarin Chinese version of the Barratt impulsiveness scale version 11 (BIS-11) was used to measure the participants’ self-reported impulsiveness [31,42]. The BIS-11 is rated on a 4-point scale with endpoints ranging from 1 (*rarely or never*) to 4 (*almost always or always*). The Mandarin Chinese version of the BIS-11 assesses three domains, namely the inability to plan and look ahead, a lack of self-control and perseverance, and novelty-seeking behavior as well as quick decision making. A higher score on the BIS-11 indicates a higher level of impulsiveness. The IABS exhibits good reliability and validity in the Taiwanese population [30]. In the present study, the BIS-11 had a Cronbach’s α of 0.78 to 0.84.

#### 2.2.3. Perceived Social Support

The family and social relationship domains of the Taiwanese quality of life questionnaire for adolescents (TQOLA) were used to measure the participants’ perceived family and peer support in the preceding month [43]. The family domain comprises seven items, the social relationship domain comprises five items, and all items are rated on a 5-point scale. A higher total domain score indicates a higher level of perceived support. The TQOLA exhibits good reliability and validity in the Taiwanese adolescent population [43,44]. The Cronbach’s α for the family and social relationship domains was 0.89 and 0.83, respectively.

#### 2.2.4. Time Spent on the Internet

The participants were asked how many minutes they spent on average using the Internet on both weekdays and weekends. The total time spent on the Internet each week was the sum of the total time spent on the Internet on weekdays and weekends.

#### 2.2.5. Twenty-Item Mandarin Chinese Version of the Center for Epidemiological Studies Depression Scale

The present study used the 20-item Mandarin Chinese version of the Center for Epidemiological Studies Depression Scale (MC-CESD) to assess the participants’ self-reported depressive symptoms [45,46]. All items were rated on a 4-point scale with endpoints from 0 (*rarely or never*) to 3 (*most or all of the time*). The MC-CES-D exhibits good reliability and validity in the Taiwanese adolescent population [47,48,49]. In the present study, the MC-CES-D had a Cronbach’s α of 0.88. An individual with a total MC-CES-D score of 29 or higher can be categorized as having severe depression [49].

#### 2.2.6. Suicidality

The 5-item suicide module of the Kiddie schedule for affective disorders and schizophrenia was used to assess the participants’ lifetime suicidal ideation and attempts [50,51]. Participants respond to each item with an answer of “yes” or “no”. Those who indicated “yes” to any of the five items were classified as having suicidality.

#### 2.2.7. Nonsuicidal Self-Injury

The *DSM-5* criterion for nonsuicidal self-injury was employed to assess the participants’ lifetime experience of nonsuicidal self-injury [41]. The participants who indicated “yes” to the question were classified as having engaged in nonsuicidal self-injury.

### 2.3. Data Analysis

Statistical analyses were performed using version 22.0 of IBM SPSS Statistics (IBM Corporation, Armonk, NY, USA). We conducted chi-square and *t*-tests to compare the demographics, time spent on the Internet, IABs, impulsiveness, perceived social support, severe depression, suicidality, and nonsuicidal self-injury of the adolescents with ADHD and the TD adolescents. To assess whether the continuous variables examined in the present study were normally distributed, the absolute values of <7 and <3 for kurtosis and skewness, respectively, were applied as criteria [52]. The results of the tests did not reveal any significant deviation.

Bivariable linear regression analysis was conducted to examine the associations of demographics, ADHD diagnosis, time spent on the Internet, impulsivity, and perceived social support with IABs. The factors that were significantly associated with IABs were included in our stepwise multivariable linear regression analysis model. In accordance with the method developed by Baron and Kenny [53], we examined the moderating effects of ADHD on the associations of independent variables with IABs. A *p*-value of <0.05 indicated statistical significance. The associations of IABs with severe depression, suicidality, and nonsuicidal self-injury were examined by performing multivariable logistic regression analysis to adjust for demographics and ADHD. The results of the analysis are presented as odds ratios (ORs) and 95% confidence intervals (CIs). The mediating effect of ADHD on the association between impulsiveness and IABs was examined using a mediation analysis.

## 3. Results

Table 1 presents the results from the comparison of the ADHD and non-ADHD groups in terms of demographics, time spent on the Internet, IABs, impulsiveness, perceived social support, severe depression, suicidality, and nonsuicidal self-injury. Compared with the adolescents without ADHD, those with ADHD obtained a higher transformed score for IABs. Relative to the adolescents without ADHD, those with ADHD reported greater impulsiveness, which is characterized by an inability to plan and look ahead as well as a lack of self-control and perseverance; the adolescents with ADHD also reported a lower level of perceived family and peer support. The group of adolescents with ADHD were more likely to have severe depression, suicidality, and nonsuicidal self-injury relative to those without ADHD. No differences in age, sex, or time spent on the Internet were identified between the adolescents with versus without ADHD.

Table 2 presents the results of our bivariable linear regression analysis, which examined the factors associated with IABs. An older age (*p* = 0.011), ADHD diagnosis (*p* = 0.049), more time spent on the Internet (*p* < 0.001), greater impulsiveness characterized by a lack of self-control and perseverance (*p* = 0.003), greater impulsiveness characterized by novelty-seeking behavior and quick decision making (*p* = 0.008), and greater perceived peer support (*p* = 0.002) were significantly associated with a higher level of IABs, whereas greater impulsiveness characterized by an inability to plan and look ahead was significantly associated with a lower level of IABs (*p* = 0.007).

The factors that were identified in the bivariable linear regression analysis as being significantly associated with IABs were included in our stepwise multivariable linear regression analysis model (Table 3). Time spent on the Internet was the first variable included in the regression analysis model (*p* < 0.001). In addition to time spent on the Internet, the following variables were identified as being significantly associated with a higher level of IABs: greater perceived peer support (*p* < 0.001), greater impulsiveness characterized by a lack of self-control and perseverance (*p* < 0.001), less inability to plan and look ahead (*p* < 0.001), and an ADHD diagnosis (*p* = 0.003; Model I). The interactions of ADHD with perceived peer support and impulsiveness were further included in a stepwise multivariable linear regression analysis model (Model II). The interaction between ADHD and perceived peer support was significantly associated with a higher level of IABs (*p* = 0.002), whereas the interaction between ADHD and impulsiveness did not exhibit this association; the positive regression coefficient (B = 0.017) indicated that the association between perceived peer support and IABs was stronger in the adolescents with ADHD than in the adolescents without ADHD. The result of the mediation analysis did not show a significant mediation of ADHD on the association between impulsiveness and IABs (B = 0.081, standard error [SE] = 0.053, *p* = 0.131).

The associations of independent factors with the four domains of IABs were also examined using the bivariable and stepwise multivariable regression analyses (Appendix A). The results of the stepwise multivariable regression analysis indicated that time spent on the Internet, greater perceived peer support, and impulsiveness characterized by a lack of self-control as well as perseverance (*p* < 0.001) were positively associated with all four domains of IABs (*p*-values: <0.001~0.019). Impulsiveness characterized by inability in planning and looking ahead was negatively associated with the two domains of IABs of Internet support (*p* < 0.001) and Internet guidance (*p* = 0.001). Boys had more Internet guidance than girls (*p* < 0.001). The associations of the diagnosis of ADHD with the four domains of IABs were nonsignificant (*p* > 0.05).

Table 4 presents the results obtained when examining the associations of IABs with severe depression, suicidality, and nonsuicidal self-injury. After the effects of age, sex, and ADHD were adjusted for, IABs were not significantly associated with severe depression, suicidality, and nonsuicidal self-injury (all *p* > 0.05).

## 4. Discussion

The present study revealed that several individual factors (e.g., an ADHD diagnosis and impulsiveness) and perceived peer support were correlated with IABs in adolescents. The association between perceived peer support and IABs was stronger in the adolescents with ADHD than that in those without ADHD. Adolescent IABs were not significantly associated with depression, suicidality, or nonsuicidal self-injury.

After the results were adjusted for time spent on the Internet, the adolescents with ADHD were found to be more likely than those without ADHD to engage in IABs. This finding supports H1. The core symptoms of ADHD, mood dysregulation, a lack of social skills, and a social stigma of ADHD all contribute to the difficulties experienced by adolescents with ADHD in developing and maintaining peer relationships in the physical world [54,55]. In contrast, the anonymous cyberworld provides adolescents with ADHD opportunities to perform IABs quickly and effectively, which may result in positive feedback and peer recognition for such adolescents and contribute to the increase in and stabilization of IABs in this population.

This study demonstrated that perceived peer support was the variable found to be associated with IABs in adolescents. This finding aligns with that of another study [11] and supports H3b. Adolescence is a developmental stage characterized by increasing independence from the family microsystem; during this period, peers play a crucial role, providing emotional support for adolescents [17]. Because the Internet is a key living space for adolescents, perceived peer support and IABs may have a reciprocal relationship, with greater perceived peer support increasing adolescents’ sense of belonging to a peer group and these adolescents then being willing to engage in IABs to help and maintain connections with others. The present study also revealed that the association between perceived peer support and IABs is stronger in adolescents with ADHD than in adolescents without ADHD. This finding supports H3d, supporting the unique role of IABs in the establishment of peer relationships for adolescents with ADHD.

The present study did not identify a significant association between perceived family support and adolescent IABs. According to Baron and Kenny [48], ADHD does not moderate the association between perceived family support and IABs in adolescents. Notably, the results of the present study do not support H3a and H3c. During the process of growing up, adolescents become increasingly independent from their families and may become less strongly influenced by their families. Adolescents develop their own attitudes and values regarding online behaviors, and such attitudes and values may diverge from those held by their families. Whether perceived family support is significantly associated with physical-world altruistic behaviors is a topic that warrants further study.

In the present study, various dimensions of impulsiveness were revealed to be associated with IABs in adolescents. This finding partially supports Hypothesis 2a. Impulsiveness characterized by a lack of self-control and perseverance was positively associated with IABs in adolescents. In the physical world, people tend to deliberate before engaging in an altruistic behavior [27,28], whereas in the anonymous cyberworld adolescents are more likely to feel safe and unrestricted and engage in IABs such as sharing information with others and providing online gaming guidelines without extensive consideration. Additionally, impulsiveness characterized by an inability to plan and look ahead was negatively associated with IABs in adolescents. This finding indicates that although the requirements for engaging in IABs are lower than those for engaging in physical-world altruistic behaviors [29,30], the ability to plan and look ahead remains essential to adolescents performing IABs. Our multivariable regression analysis revealed that IABs were not significantly associated with impulsiveness characterized by novelty-seeking behavior or quick decision making, suggesting that novelty-seeking behavior does not play a crucial role in motivating adolescents to engage in IABs. The results of the present study do not support the moderating effect of ADHD on the association between impulsiveness and IABs in adolescents, indicating that the association of impulsiveness with IABs is not limited to adolescents with ADHD.

In the present study, IABs were not significantly associated with depression, suicidality, and nonsuicidal self-injury in adolescents. These results do not support H4 and contradict the results of other studies on IABs [5,13]. A longitudinal intervention study demonstrated that physical-world altruistic behaviors play a crucial role in promoting adolescents’ academic achievement and mental health [36]. Feng and Guo discovered that physical-world altruistic behaviors are conducive to the realization of self-value and the improvement of well-being [37]. Given that the Internet is a key aspect of the lives of adolescents, the relationship between IABs and mental health in the adolescent population warrants further study.

### Limitations

The present study has several limitations: First, we recruited adolescents with ADHD from outpatient clinics who were currently receiving pharmacological or psychological therapies. Their parents also received guidance from child psychiatrists to improve parent–child interactions. Adolescents without ADHD were recruited through an online advertisement; their Internet use behaviors might not be the same as of those who did not participate in this study. This difference in recruitment methods could have introduced sampling bias into our research. A topic that warrants further study is whether the results of the present study can be generalized to adolescents with ADHD who do not visit outpatient clinics for medical help and adolescents who are not recruited through online advertisements. Moreover, the education systems in East Asian regions such as China, Taiwan, Korea, and Japan are strongly influenced by ideas from Confucianism, which eschews play and emphasizes hard work, effort, persistence, self-cultivation, and discipline in one’s studies [56]. Adolescents under academic pressure have a high tendency to seek pleasure from online activities, especially online gaming; thus, problematic Internet use is a significant adolescent health issue in these East Asian regions [57,58]. Whether the results of this study examining adolescent IABs can be generalized to adolescents from other cultures and societies warrants further study. Second, the cross-sectional design of the present study prevented us from clarifying the temporal associations of IABs with other variables; therefore, readers must interpret the results of the present study with caution to avoid drawing conclusions regarding reverse causation. Further follow-up studies are needed to examine the temporal relationships of individual and environmental factors with IABs. Third, this study collected participants’ self-reported data; therefore, single-rater and recall biases could not be fully controlled for. To address this limitation, future studies should examine potential information bias due to social desirability. Obtaining information from multiple sources may confirm the accuracy of the data. Fourth, we did not examine the Internet activities that adolescents engaged in. Adolescents may have various IABs when they engage in different Internet activities.

## 5. Conclusions

The present study highlights the complex interplay between IABs and ADHD, impulsiveness, and perceived peer support in adolescents. With the Internet becoming a key space for modern adolescents to engage in interpersonal interaction, medical professionals, educators, and parents must develop an in-depth understanding of IABs in adolescents. The Internet is an important living space for adolescents with ADHD; they can explore themselves, develop their self-identity, and enhance their peer support [10]. Previous studies have focused on how Internet use negatively affects adolescents’ daily lives and health [6,7,8,9]; however, adolescents with ADHD should also be recognized as engaging in positive behaviors on the Internet. The present study is the first to examine the IABs of adolescents with a focus on adolescents with ADHD and found that adolescents with ADHD reported greater IABs compared with those without ADHD. This study also demonstrates the moderation of ADHD on the association between perceived peer support and IABs, indicating a strong correlation between interpersonal interactions in the real world and online among adolescents with ADHD. Given that individuals with ADHD are at a disadvantage in real-world interpersonal interactions [16], health professionals and parents need to help adolescents with ADHD to extend altruistic behaviors on the Internet into the real world, thereby developing good interpersonal relationships.

The present study also provides insights into the complex relationships of IABs with various domains of impulsiveness. Further studies need to investigate how various domains of impulsiveness have different impacts on the developments of IABs. Given that an inability in planning and looking ahead was negatively associated with IABs, assisting adolescents in developing the ability to plan and look ahead is necessary to improve IABs. Moreover, although impulsiveness characterized by a lack of self-control and perseverance was positively associated with IABs, health professionals and parents need to help adolescents adapt this impulsive trait so that IABs are consistently present and contextualized. Although this study did not find significant associations of IABs with severe depression, suicidality, and nonsuicidal self-injury, whether IABs can help adolescents recognize their self-worth and experience active emotions warrants further study.

## Figures and Tables

**Table 1 behavsci-14-00433-t001:** Demographics, time spent on the Internet, Internet altruistic behaviors, impulsiveness, social support, depression, suicidality, and nonsuicidal self-injury in adolescents with and without ADHD.

	Non-ADHD (*n* = 173)	ADHD (*n* = 176)	*t* or χ^2^	*p*
Age (years), mean (SD)	13.7 (2.1)	13.7 (2.1)	−0.149	0.881
Sex, *n* (%)				
Girls	32 (18.5)	32 (18.2)	0.006	0.939
Boys	141 (81.5)	144 (81.8)		
Time spent on the Internet (hours per week), mean (SD)	17.7 (18.9)	19.6 (16.9)	−1.014	0.311
Internet Altruistic Behaviors, Mean (SD)				
Support	13.5 (5.1)	14.5 (5.1)	−1.859	0.064
Guidance	8.2 (2.7)	8.8 (3.4)	−1.901	0.058
Sharing	10.0 (3.4)	10.5 (3.5)	−1.393	0.164
Reminding	6.6 (2.3)	7.0 (2.4)	−1.648	0.100
Transformed score	-0.1 (1.0)	0.1 (1.0)	−1.952	0.049
Impulsiveness on the BIS-11, Mean (SD)				
Inability in planning and looking ahead	20.4 (4.0)	22.4 (4.2)	−4.620	<0.001
Lack of self-control and perseverance	23.2 (4.4)	24.6 (4.8)	−2.993	0.003
Novelty seeking and quick in decision making	15.0 (2.8)	15.5 (3.7)	−1.359	0.175
Perceived peer support, mean (SD)	19.4 (3.2)	17.9 (3.7)	3.855	<0.001
Perceived family support, mean (SD)	28.2 (4.9)	26.8 (5.6)	2.439	0.015
Significant depression, *n* (%)	9 (5.2)	30 (17.0)	12.328	<0.001
Suicidality, *n* (%)	23 (13.3)	43 (24.4)	7.057	0.008
Nonsuicidal self-injury, *n* (%)	12 (6.9)	27 (15.3)	6.208	0.013

ADHD: attention-deficit/hyperactivity disorder; SD: standard deviation.

**Table 2 behavsci-14-00433-t002:** Factors related to Internet altruistic behaviors: bivariable linear regression analysis.

	Internet Altruistic Behaviors	
	B (SE)	*p*
Age	0.064 (0.025)	0.011
Sex ^a^	0.224 (0.138)	0.138
ADHD	0.208 (0.107)	0.049
Time spent on the Internet	0.011 (0.003)	<0.001
Inability in planning and looking ahead	−0.034 (0.013)	0.007
Lack of self-control and perseverance	0.034 (0.011)	0.003
Novelty seeking and quick in decision making	0.043 (0.016)	0.008
Perceived family support	0.001 (0.010)	0.913
Perceived peer support	0.046 (0.015)	0.002

^a^: Girls were used as the reference. ADHD: attention-deficit/hyperactivity disorder.

**Table 3 behavsci-14-00433-t003:** Factors related to Internet altruistic behaviors: stepwise multivariable linear regression analysis.

Variables	Internet Altruistic Behaviors			
	Model I		Model II	
	B (SE)	*p*	B (SE)	*p*
Time spent on the Internet	0.010 (0.003)	<0.001	0.010 (0.003)	<0.001
Perceived peer support	0.054 (0.015)	<0.001	0.044 (0.015)	0.004
Lack of self-control and perseverance	0.058 (0.012)	<0.001	0.058 (0.012)	<0.001
Inability in planning and looking ahead	−0.062 (0.014)	<0.001	-0.063 (0.014)	<0.001
ADHD	0.305 (0.102)	0.003	–	–
ADHD × perceived peer support	–	–	0.017 (0.005)	0.002
F	13.580	<0.001	13.865	<0.001
Adjusted R^2^	0.153		0.156	

ADHD: attention-deficit/hyperactivity disorder.

**Table 4 behavsci-14-00433-t004:** Associations of Internet altruistic behaviors with severe depression, suicide, and nonsuicidal self-injury: multivariable logistic regression analysis.

	Significant Depression		Suicide		Nonsuicidal Self-Injury	
	OR (95% CI)	*p*	OR (95% CI)	*p*	OR (95% CI)	*p*
Gender	0.710 (0.311–1.618)	0.415	1.163 (0.562–2.406)	0.685	1.620 (0.602–4.357)	0.339
Age	1.141 (0.974–1.336)	0.102	1.146 (1.009–1.301)	0.036	1.048 (0.895–1.227)	0.563
ADHD	3.765 (1.720–8.242)	0.001	2.105 (1.198–3.698)	0.010	2.445 (1.191–5.020)	0.015
IABs	1.021 (0.728–1.432)	0.904	1.037 (0.794–1.355)	0.789	0.971 (0.694–1.357)	0.861

ADHD: attention-deficit/hyperactivity disorder; CI: confidence interval; IABs: Internet altruistic behaviors; and OR: odds ratio.

## Data Availability

The data are available upon reasonable request to the corresponding authors.

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
