# Peer review of "Internet Altruistic Behaviors in Adolescents: Roles of Attention-Deficit/Hyperactivity Disorder, Impulsiveness, and Perceived Social Support"

_behavsci, 2024, doi:10.3390/bs14060433_

Round 1

Reviewer 1 Report

Comments and Suggestions for Authors

Thank you for the opportunity to review your article which investigates an interesting and very current topic.

The article is clear and coherent. The supporting scientific literature is not abundant but sufficient to substantiate the topic. The hypotheses are clearly presented as well as the method followed and the statistical analysis carried out (chi-square, t-test, multivariable linear regression analysis and multivariable logistic regression analysis).

The results are clearly presented and discussed referring to the expected hypotheses.

The limitations of the study are adequately described.

The only note I would like to point out is that it would be interesting to add practical implications or directions for future research on the topic.

Furthermore, the results are adequately addressed in the discussion in relation to the hypotheses and research questions. However, the conclusion is very brief and general, it could be improved by describing in a concise and specific way what are the main results discussed previously and what their implications in terms of impact and future research.

Author Response

Dear Reviewer 1

We appreciated your valuable comments. As discussed below, we have revised our manuscript with underlines based on your suggestions. Please let us know if we need to provide anything else regarding this revision.

Comment 1

The only note I would like to point out is that it would be interesting to add practical implications or directions for future research on the topic. The conclusion is very brief and general, it could be improved by describing in a concise and specific way what are the main results discussed previously and what their implications in terms of impact and future research.

Response

Thank you for your comment. We revised the Conclusion section and added practical implications and directions for future research on the topic into the revised manuscript. Please refer to line 398-427.

The present study highlights the complex interplay between IABs and ADHD, impulsiveness, and perceived peer support in adolescents. With the Internet becoming a key space for modern adolescents to engage in interpersonal interaction, medical professionals, educators, and parents must develop an in-depth understanding of IABs in adolescents. The Internet is an important living space for adolescents with ADHD; they can explore themselves, develop their self-identity, and enhance their peer support [10]. Previous studies have focused on how Internet use negatively affects adolescents’ daily lives and health [6–9]; however, adolescents with ADHD should also be recognized for positive behaviors in the Internet. The present study is the first to examine the IABs of adolescents with a focus on adolescents with ADHD and found that adolescents with ADHD reported greater IABs compared with those without ADHD. This study also demonstrates the moderation of ADHD on the association between perceived peer support and IABs, indicating a strong correlation between the interpersonal interactions in the real world and online among adolescents with ADHD. Given that individuals with ADHD are at a disadvantage in real-world interpersonal interactions [16], health professionals and parents need to help adolescents with ADHD to extend altruistic behaviors in the Internet into the real world, thereby developing good interpersonal relationships.

The present study also provides insights into the complex relationships of IABs with various domains of impulsiveness. Further studies need to investigate how various domains of impulsiveness have different impacts on the developments of IABs. Given that an inability in planning and looking ahead was negatively associated with IABs, assisting adolescents in developing the ability in planning and looking ahead is necessary to improve IABs. Moreover, although impulsiveness characterized by lack of self-control and perseverance was positively associated with IABs, health professionals and parents need to help adolescents adapt this impulsive trait so that IABs are consistently present and contextualized. Although this study did not find the significant associations of IABs with severe depression, suicidality, and nonsuicidal self-injury, whether IABs can help adolescents recognize their self-worth and experience active emotions warrants further study.”

Reviewer 2 Report

Comments and Suggestions for Authors

The manuscript seems fine. I think is a good correlational study.

I would like to ask for the referred instrument:

First, you comment in your article the issue about generalization between typo of patients. I just want to know your opinion about what is the level of generalization between these findings specific for taiwanese-chinese adolescents and adolescents from other cultures?

Another relevant question I would like you to answer is what type of transformation is done for the Transformed Score in IABs. Is it something new for this study? I have doubts about the relevance of these findings, taking into account that support, guidance, sharing and reminding were not significant.

Do you consider the topic original or relevant in the field? Does it address a specific gap in the field?

What does it add to the subject area compared with other published material?

What specific improvements should the authors consider regarding the methodology? What further controls should be considered?

Author Response

Dear Reviewer 2

We appreciated your valuable comments. As discussed below, we have revised our manuscript with underlines based on your suggestions. Please let us know if we need to provide anything else regarding this revision.

Comment

  1. You comment in your article the issue about generalization between typo of patients. I just want to know your opinion about what is the level of generalization between these findings specific for taiwanese-chinese adolescents and adolescents from other cultures?

Response

Thank you for your comment. We added discussion regarding the differences in educational expectations between East Asia and other cultures and their influences on the generalization of the results of this study into Discussion section. Please refer to line 378-385.

“…the education system in East Asian regions such as China, Taiwan, Korea and Japan and is strongly influenced by ideas in Confucianism, which eschews play and emphasizes hard work, effort, persistence, self-cultivation, and discipline in one’s studies [56]. Adolescents under academic pressure have a high tendency to seek pleasure from online activities, especially online gaming; thus, problematic Internet use is a significant adolescent health issue in these East Asian regions [57,58]. Whether the results of this study examining adolescent IABs can be generalized to the adolescents from other cultures and societies warrants further study.

Comment

  1. Another relevant question I would like you to answer is what type of transformation is done for the Transformed Score in IABs. Is it something new for this study? I have doubts about the relevance of these findings, taking into account that support, guidance, sharing and reminding were not significant.

Response

Thank you for your comment. We transformed the scores for the four subscales of the IABS to extract a factor score of IABs through factor analysis (principal components method). In the revised manuscript, we also examined the factors related to the four domains of IABS, named support, guidance, sharing and reminding and added the results into Results section and Supplementary tables.

  • Because of the high correlation between subscale scores (Pearson’s correlation coefficients: 0.526~0.741), the scores for the four subscales of the IABS were transformed through factor analysis (principal components method) to extract a factor score that represented IABs.” Please refer to line 174-177.
  • The associations of independent factors with the four domains of IABs were also examined using the bivariable and stepwise multivariable regression analysis (Supplementary Tables 1 and 2). The results of stepwise multivariable regression analysis indicated that time spent on the Internet, greater perceived peer support, impulsiveness characterized by a lack of self-control and perseverance (p < 0.001) were positively associated with all four domains of IABs (p values: <0.001~0.019). Impulsiveness characterized by inability in planning and looking ahead was negatively associated with the two domains of IABs of Internet support (p <0.001) and Internet guidance (p = 0.001). Boys had more Internet guidance than girls (p < 0.001). The associations of the diagnosis of ADHD with the four domains of IABs were nonsignificant (p > 0.05).” Please refer to line 284-293.

Comment

  1. Do you consider the topic original or relevant in the field? Does it address a specific gap in the field?
  2. What does it add to the subject area compared with other published material?

Response

Thank you for your comment. The present study investigated several topics pertaining to adolescent IABs. First, most studies have examined the IABs of college students, with only one study examining the IABs of middle school students. Notably, no study has examined the IABs of adolescents with ADHD. Given that adolescents with ADHD have a higher tendency to have problematic Internet use compared with those without ADHD, examining IABs in adolescents with ADHD can provide knowledge. Moreover, according to ecological theory, IABs are the results of interactions between individuals and their environments. The present study examined several individual factors (i.e., time spent on the Internet, the diagnosis of ADHD, and the characteristics of impulsiveness) and environmental factors (i.e., perceived family and peer support). We believe that the results of this study addressed specific gaps in the field and add new knowledge to the area of IAB study. We described these topics newly investigated in the present study in Introduction section.

Comment 5

What specific improvements should the authors consider regarding the methodology? What further controls should be considered?

Response

Thank you for your comment. The methods of recruiting participants, study design, sources of information, and examined factors can be improved in further studies.

  • First, we recruited adolescents with ADHD from outpatient clinics who were currently receiving pharmacological or psychological therapies. Their parents also received guidance from child psychiatrists to improve parent-child interactions. Adolescents without ADHD were recruited through an online advertisement; their Internet use behaviors might not the same with those who did not participate in this study. This difference in recruitment methods could have introduced sampling bias into our research. A topic that warrants further study is whether the results of the present study can be generalized to adolescents with ADHD who do not visit outpatient clinics for medical help and adolescents who are not recruited through online advertisements.” Please refer to line 369-378.
  • “Second, the cross-sectional design of the present study prevented us from clarifying the temporal associations of IABs with other variables. Therefore, readers must interpret the results of the present study with caution to avoid drawing conclusions regarding reverse causation. Further follow-up studies are needed to determine the temporal relationships of the factors examined in this study with IABs.” Please refer to line 386-390.
  • “…all data were self-reported by the participants; therefore, the researchers could not fully control for single-rater and recall biases…Obtaining information from multiple sources may confirm the accuracy of the data.” Please refer to line 390-394.
  • Fourth, we did not examine the Internet activities that adolescents engaged in. Adolescents may have various IABs when they engage in different Internet activities.” Please refer to line 394-396.

Reviewer 3 Report

Comments and Suggestions for Authors

This study explored the relationship between Internet altruistic behaviors and several other variables including ADHD, impulsivity and perceived social support among adolescents. It addressed an interesting topic of adolescents’ prosocial online behaviours. I have several comments as below.

1.     The authors need to further explain why they selected the variables with a clear rationale or theoretical model.

2.     This study investigated the related factors with IAB and hypothesised that higher IAB is related with ADHD, impulsivity and social support. I am confused why this is important. What’s the practical research implication for this? Why you explored the interaction?

3.     This study also investigated IAB and depression, suicide and self-injury. The meaningfulness of this also requires further justification. How could positive altruistic behaviors online related to these mental health problems?

4.     The interactions need to be further explained with more details of simple slope analysis and figures.

5.     It seems suitable to use SEM to include all the variables in a model rather than analyse them in two steps. Mediation and moderation models tested in SPSS PROCESS are also recommended for the variables.

6.     In discussion chapter, the authors need to address more about the theoretical implication of the interactions identified in this study, and focus on the clinical implications of the findings.

7.     Please further check your grammar errors and vague expressions in the manuscript. For example: in line 71, “Furthermore, online games are a main leisure activity for adolescents with ADHD”.

Comments on the Quality of English Language

Please check your grammar errors. 

Author Response

Dear Reviewer 3

We appreciated your valuable comments. As discussed below, we have revised our manuscript with underlines based on your suggestions. Please let us know if we need to provide anything else regarding this revision.

Comment

  1. The authors need to further explain why they selected the variables with a clear rationale or theoretical model.
  2. This study investigated the related factors with IAB and hypothesised that higher IAB is related with ADHD, impulsivity and social support. I am confused why this is important. What’s the practical research implication for this? Why you explored the interaction?

Response

Thank you for your comment. We added the explanations for the reasons of selecting the variables into examination in Introduction section.

  • ADHD: “According to ecological systems theory [17], IABs result from the interactions between individuals and their environment. Given that adolescents with ADHD exhibit the psychological characteristics of boredom proneness, rapid habituation to repeated positive reinforcement, and a high propensity for seeking fun, the Internet provides a conducive space for them to pursue pleasurable experiences [18,19]. The Internet also gives adolescents with ADHD opportunities to develop social relationships and gain peer recognition, which are more difficult for them to achieve through face-to-face interactions [20]. Furthermore, Internet gaming is an important leisure activity for adolescents with ADHD [21], and the Internet offers numerous opportunities to share gaming tips with others. Studies have demonstrated that ADHD is one of risk factors for developing problematic Internet use [22–24]. Studies have also focused on the negative effects of problematic Internet use on mental health and social skills among adolescents [25,26]. However, the IABs of adolescents with ADHD remain unexplored.” Please refer to line 66-78.
  • Impulsiveness: “Second, scholars have proposed that physical-world altruistic behaviors are the product of rational thinking [27] and inner responsibility [28]. Comparatively, the requirements for engaging in online IABs are less strict than those for engaging in physical-world altruistic behaviors; therefore, individuals are more likely to engage in online IABs without careful consideration [29,30]. Whether the psychological trait of impulsiveness contributes to adolescents’ IABs is unclear….Given that impulsivity is a core symptom of ADHD, whether ADHD moderates or mediates the association between impulsiveness and adolescent IAB must be clarified.” Please refer to line 82-93.
  • Family and peer support: “Third, families and peers are important sources of support for adolescents. Studies have reported that parental emotional warmth [12,33] and peer support [11] are positively associated with IABs. Some scholars hypothesized that individuals experience active emotions when they receive social support from their environment and are then willing to engage in altruistic behaviors [11]. However, adolescents with ADHD experience more conflict with their parents [34] and peer rejection [35] relative to those without ADHD; thus, the relationship between perceived social support and IABs may be different for adolescents with ADHD versus those without ADHD.” Please refer to line 100-107.

Comment

  1. This study also investigated IAB and depression, suicide and self-injury. The meaningfulness of this also requires further justification. How could positive altruistic behaviors online related to these mental health problems?

Response

Thank you for your comment. We introduced the reasons for examining the associations of IABs with depression, suicide and self-injury based on the results of previous studies. Please refer to line 115-118.

“…studies have discovered that both physical-world altruistic behaviors [36–38] and IABs [5,13] have positive associations with mental health. Altruistic behaviors have been proposed to help individuals recognize their self-worth and experience active emotions, thereby improving their mental health [33].

Comment

  1. The interactions need to be further explained with more details of simple slope analysis and figures.

Response

Thank you for your comment. We added thefigure of slope analysis in Supplementary Figure 1.

The results of slope analysis are shoewn in Supplementary Figure 1.” Please refer to line 280-281.

Comment

  1. It seems suitable to use SEM to include all the variables in a model rather than analyse them in two steps. Mediation and moderation models tested in SPSS PROCESS are also recommended for the variables.

Response

Thank you for your comment. In the revised manuscript, we added the result of mediation analysis for the mediating effect of ADHD on the association between impulsiveness and IABS.

  • Given that impulsivity is a core symptom of ADHD, whether ADHD moderates or mediates the association between impulsiveness and adolescent IAB must be clarified.” Please refer to line 91-93.
  • H2c: ADHD mediates the association between impulsiveness and IABs in adolescents.” Please refer to line 98-99.
  • The mediating effect of ADHD on the association between impulsiveness and IABs was examined using a mediation analysis.” Please refer to line 238-240.
  • The result of the mediation analysis did not show a significant mediation of ADHD on the association between impulsiveness and IABs (B = 0.081, standard error [SE] = 0.053, p = 0.131).” Please refer to line 281-283.

Comment

  1. In discussion chapter, the authors need to address more about the theoretical implication of the interactions identified in this study, and focus on the clinical implications of the findings.

Response

Thank you for your comment. We added practical implications and directions for future research on the topic into the revised manuscript. Please refer to line 398-427.

The present study highlights the complex interplay between IABs and ADHD, impulsiveness, and perceived peer support in adolescents. With the Internet becoming a key space for modern adolescents to engage in interpersonal interaction, medical professionals, educators, and parents must develop an in-depth understanding of IABs in adolescents. The Internet is an important living space for adolescents with ADHD; they can explore themselves, develop their self-identity, and enhance their peer support [10]. Previous studies have focused on how Internet use negatively affects adolescents’ daily lives and health [6–9]; however, adolescents with ADHD should also be recognized for positive behaviors in the Internet. The present study is the first to examine the IABs of adolescents with a focus on adolescents with ADHD and found that adolescents with ADHD reported greater IABs compared with those without ADHD. This study also demonstrates the moderation of ADHD on the association between perceived peer support and IABs, indicating a strong correlation between the interpersonal interactions in the real world and online among adolescents with ADHD. Given that individuals with ADHD are at a disadvantage in real-world interpersonal interactions [16], health professionals and parents need to help adolescents with ADHD to extend altruistic behaviors in the Internet into the real world, thereby developing good interpersonal relationships.

The present study also provides insights into the complex relationships of IABs with various domains of impulsiveness. Further studies need to investigate how various domains of impulsiveness have different impacts on the developments of IABs. Given that an inability in planning and looking ahead was negatively associated with IABs, assisting adolescents in developing the ability in planning and looking ahead is necessary to improve IABs. Moreover, although impulsiveness characterized by lack of self-control and perseverance was positively associated with IABs, health professionals and parents need to help adolescents adapt this impulsive trait so that IABs are consistently present and contextualized. Although this study did not find the significant associations of IABs with severe depression, suicidality, and nonsuicidal self-injury, whether IABs can help adolescents recognize their self-worth and experience active emotions warrants further study.”

Comment

  1. Please further check your grammar errors and vague expressions in the manuscript. For example: in line 71, “Furthermore, online games are a main leisure activity for adolescents with ADHD”.

Response

Thank you for your comment. We revised this sentence into “Internet gaming is an important leisure activity for adolescents with ADHD…” Please refer to line 73-74.

We also invited an English-native editor to edit this manuscript.  Attached please find that certificate of edition.
